# Cell Membrane-Coated Nanoparticles for Precision Medicine: A Comprehensive Review of Coating Techniques for Tissue-Specific Therapeutics

**DOI:** 10.3390/ijms25042071

**Published:** 2024-02-08

**Authors:** Andrés Fernández-Borbolla, Lorena García-Hevia, Mónica L. Fanarraga

**Affiliations:** 1The Nanomedicine Group, Institute Valdecilla-IDIVAL, 39011 Santander, Spain; andres.fernandezbor@unican.es (A.F.-B.); lorena.garciahevia@unican.es (L.G.-H.); 2Molecular Biology Department, Faculty of Medicine, Universidad de Cantabria, 39011 Santander, Spain

**Keywords:** nanomedicine, biomimicry, biomimetic nanoparticle, targeted drug delivery, homotypic targeting, nanoparticle coating

## Abstract

Nanoencapsulation has become a recent advancement in drug delivery, enhancing stability, bioavailability, and enabling controlled, targeted substance delivery to specific cells or tissues. However, traditional nanoparticle delivery faces challenges such as a short circulation time and immune recognition. To tackle these issues, cell membrane-coated nanoparticles have been suggested as a practical alternative. The production process involves three main stages: cell lysis and membrane fragmentation, membrane isolation, and nanoparticle coating. Cell membranes are typically fragmented using hypotonic lysis with homogenization or sonication. Subsequent membrane fragments are isolated through multiple centrifugation steps. Coating nanoparticles can be achieved through extrusion, sonication, or a combination of both methods. Notably, this analysis reveals the absence of a universally applicable method for nanoparticle coating, as the three stages differ significantly in their procedures. This review explores current developments and approaches to cell membrane-coated nanoparticles, highlighting their potential as an effective alternative for targeted drug delivery and various therapeutic applications.

## 1. Introduction

Nanoencapsulation for in vivo administration provides numerous benefits, such as enhancing effectiveness and safety by protecting the substances from degradation or elimination [1,2]. This technique contributes to increased absorption and improved bioavailability, optimizing distribution, and extending circulation time while simultaneously reducing toxicity [2,3]. Some nanomaterials offer advantages such as enhanced solubility and loading capacity, improved delivery efficiency, and protection from degradation due to the stability provided by the nanocarriers [1,2,3]. However, nanoparticle delivery has many limitations. Nanocarriers are very prone to interact with biomolecules in the bloodstream, creating the so-called “biocorona” [4], which results in recognition by the immune system [5]. Upon arrival to the target cells, many nanocarriers are trapped in endocytic vesicles and end up being degraded by lysosomes, diminishing the drug delivery efficiency [6].

Recent studies suggest that nanocarriers show an average efficiency of delivering to the desired target of less than 1% [7,8], leaving space for a significant improvement in targeted delivery. As a result, nanoparticles coated with cell membranes have been proposed as a way to address these problems, as they show a combination of the advantages present in natural nanomaterials such as cell membrane-derived nanomaterials, and artificial nanocarriers, such as the aforementioned polymeric or inorganic nanocarriers [9,10,11,12].

Cell membrane-coated nanoparticles are biomimetic nanoparticles that are constituted by a cell membrane cover and synthetic nanoparticles [5]. They offer several advantages over bare nanomaterials, such as increased biocompatibility, due to the similarity of biological membranes to cellular materials, reducing the risk of immune system rejection [13]. The presence of biological membranes enhances biodistribution by guiding nano-vectored materials to target cells, utilizing membrane receptors recognizable by the target cells. This aspect represents a significant area of study, applicable to immune system cells [13], central nervous system [14], as well as a large number of cancer cells (Table 1). Additionally, coated nanocarriers demonstrate improved drug release control and efficiency, as the biological membranes can degrade or fuse with target cells, releasing the drug at the desired location [9,15]. Specifically, the biological camouflage provided by these membranes protects nanoparticles from the body’s defense systems, extending their lifespan and reducing the risk of premature elimination [13,15,16,17,18,19,20]. The ability to target particles to specific cells, facilitated by the presence of receptors on biological membranes, is a key advantage that positions nanomaterials coated with biological membranes as a promising option for targeted delivery.

As this pioneering methodology is still in its nascent stages, our study aims to comprehensively review the recent advancements in this technology. Specifically, we delve into various studies conducted to date, focusing on elucidating the techniques employed for obtaining cell membrane fragments. We provide detailed insights into the processes involved in isolating these membranes and coating nanoparticles with them. The ultimate goal of this review is to examine technology which has been developed recently to generate cell membrane-coated nanoparticles, showcasing their potential for achieving tissue-specific targeting. This review aims to clearly outline the significance of the study within the broader context of this emerging field.

## 2. General Procedure

To obtain cell membrane-coated nanoparticles three pivotal and indispensable steps must be undertaken. These steps encompass the cell lysis and fragmentation of the membranes, the isolation of these membrane fragments, and the coating of the selected nanocarriers (Figure 1).

The choice of materials for each of these crucial steps depends on the specific tissue being targeted and the nature of the treatment under investigation. The selection is tailored to optimize compatibility with the intended biological environment and enhance the efficacy of the experimental approach.

## 3. Membrane Donor Cells

The selection of a specific cell type is contingent upon the target tissue or application. Typically, cancer cells are employed to specifically target the corresponding cancerous tissue, while white or red blood cells may be used for applications with less specific targets. Most of these cell types were employed to facilitate the precise targeting of nanoparticles to specific tissues. However, some of these cells served a dual purpose by inducing immune stimulation against cancer.

M any different cell types have been used for nanoparticle membrane coating (Table 1). Notably, a range of cancer lines has been used, including cervical and ovarian cancers [21,22,23,24], multiple myeloma [25], melanoma [12,26,27,28,29,30,31,32], leukemia [23,33,34,35,36,37,38,39,40,41,42,43,44], breast cancer [6,37,40,45,46,47,48,49,50,51,52,53,54,55,56], neuroblastoma [79], colon carcinoma [23,57], head and neck squamous cell carcinoma [58,59,60,61], lung cancer [54,62], glioma [63,64], glioblastoma [65,66], prostate cancer [67], and liver cancer [68]. Furthermore, beyond cancer cells, a multitude of non-cancer cells has also been utilized, such as leukocytes [92,93], macrophages [92,94,95,96,97,98,99], erythrocytes [19,29,46,48,80,81,82,83,84,85,86,87,88,89,90,91], dendritic cells [100], neutrophils [89,101,102,103,104], mesenchymal stem cells [74,75,76,77,78], platelets [48,86,87,93,105,106,107], fibroblasts [49,69], embryonic kidney cells [70], vaginal endothelial cells [71], neural stem cells [72], microglial cells [66], and keratinocytes [73]. 

Cervical and ovarian cancer cells were used to favor the cytosolic delivery of cargo inside living cells [21] or for homologous targeting [22]. Multiple myeloma cells were chosen to target their equivalent counterparts, ensuring specificity in cargo delivery [25]. In the case of melanoma cells, their use was geared towards promoting the delivery and internalization of therapeutic or antigenic materials [12], or for photoimmunotherapy [26]. Leukemia cells were employed to deliver cargo into leukemia cells [34] or were genetically modified to express a protein that can specifically target a tissue [36]. Neuroblastoma cells were employed for their capacity to capture neurotoxins effectively [79]. Breast cancer cells were used to target homologous cells and deliver cargo [6]. Similarly, colon carcinoma [57] head and neck squamous cell carcinoma [58], lung cancer [54], glioma [63,64], glioblastoma [65,66], prostate cancer [67], and liver cancer [68] cells were selected for homologous targeting, ensuring precision in cargo delivery to specific tissues. 

In the case of non-cancer cells, leukocytes were harnessed for their capacity to target specific tissues effectively [92]. Erythrocytes were used to target cancer tissues, due to their elasticity and capacity to diffuse into the tumor extracellular matrix [81]. Dendritic cells were employed to promote tumor immune effects [100]. Vaginal endothelial cells were used to protect the cells from a toxin [71]. Neural stem cells were used to cross the blood-brain barrier and specific targeting [72]. Neutrophils [101], mesenchymal stem cells [74], fibroblasts [49,69], embryonic kidney cells [70], microglial cells [66], and keratinocytes [73] were also used for specific targeting.

Some investigations opted to combine membranes from different cells so that the coated nanoparticles benefited from the characteristics of both types of source cells. When hybrid membrane-coated nanoparticles were developed by combining two cell types, leukocytes were chosen to mitigate immune recognition [93], platelets were selected for their notable ability to bind to cancer cells [93], and erythrocytes due to their long circulation times [48] and immune-evasion capability [29]. Additionally, breast cancer cells [46,48], were incorporated in hybrid membrane coating to ensure precise targeting of homologous cells.

## 4. Fragmentation of Cell Membranes

The initial crucial step in the preparation of cell membrane-coated nanoparticles involves the obtention of purified cell membrane fragments. Various techniques are employed to produce these membrane fragments, with hypotonic lysis, homogenization, freeze-thaw, and sonication emerging as the most commonly utilized methods (Figure 2). Often, these methods are used together to enhance results, such as combining hypotonic lysis, homogenization, and freeze-thaw for improved outcomes.

### 4.1. Hypotonic Lysis

Most researchers employed hypotonic lysis in their studies [6,12,19,21,22,23,24,25,26,27,29,30,31,32,33,34,35,36,38,39,41,42,43,44,45,48,49,50,51,54,55,56,57,58,60,61,62,63,64,65,66,67,69,70,71,72,73,74,75,76,77,79,81,82,84,87,88,89,90,91,92,93,95,97,98,100,101,102,103,104]. This lysis method involves resuspending the utilized cells in a hypotonic solution containing low concentrations of salts and protease or phosphatase inhibitors. Several authors used a hypotonic lysis buffer with 20 mM Tris-HCl pH 7.5, 10 mM KCl, 2 mM MgCl_2_, and 1 EDTA-free mini protease inhibitor tablet per 10 mL of solution [12,23,25,34,44,45,51,54,56,58,67,72,95]. Parodi et al. drew upon the use of the same salts as Qu et al., but adding 25 mM of sucrose and using PMSF and trypsin-chymotrypsin inhibitors [92]. A similar buffer was used by Li et al. (Tris-HCl, 20 mM KCl, 2 mM MgCl_2,_ and EDTA-free-microprotease inhibitor) [62]. Other authors utilized Tris-HCl, sucrose, and D-mannitol in combination with phosphatase and protease inhibitor cocktails [27,30,31,71]. In contrast, a handful of researchers used these components along with EGTA (IB-1 buffer) [38,50,98,101,102], while Nie et al. used this IB-1 buffer with 0.5% (*w*/*v*) BSA [6]. Ma et al. opted for the commercial RIPA Lysis Buffer (50 mM Tris-HCl pH 7.6, 150 mM NaCl, 1% NP-40, 0.5% sodium deoxycholate, 0.1% SDS) in addition to a protein inhibitor cocktail [100]. Others used simpler Tris-HCl lysis buffers, such as Bu et al. (50 mM Tris-HCL pH 7.4) [73], Ma et al. and Zou et al. (10 mM Tris and 10 mM MgCl_2_ EDTA free protease inhibitor) [63,77], or Liu et al. (Tris-HCl pH 7.4, 10 mM MgCl2, 1× PMSF, 0.2 mM EDTA and phosphatase inhibitor cocktail) [69].

Other variations in hypotonic buffers were observed, such as the use of a hypotonic buffer with 0.25X PBS [61,76,81,82,88,90,91] containing a protease inhibitor cocktail [21] or PMSF [46]. PBS was also used in combination with EDTA-2Na [87]. Jiang et al. and Rao et al. used Hepes B buffer (10 mM Hepes, 5 mM MgCl_2_, 1 mM EDTA, 1 mM DTT, 10 mM KCl, pH 7.6) mixed with protease inhibitor tablets [48,93], and Li et al. used a similar homogenization medium with 20 mM HEPES-NaOH, 1 mM EDTA, and 0.25 M of sucrose with PMSF [22]. As an alternative to EDTA-containing buffers, Wang et al. and Park et al., employed EGTA in combination with a phosphatase and protease inhibitor [35,79]. Jiang et al. opted for a NaHCO_3_ based buffer (1 mM NaHCO_3_, 0.2 mM EDTA∙2Na, 1 mM PMSF and 1 × PIC in H_2_O) [39], while Du et al. used a similar buffer [64]. Li et al. used double distilled water [103]. Some articles did not specify the exact buffer composition but indicated the use of a low-osmotic lysis buffer containing membrane protein extraction reagents and PMSF [26]. Wu et al. subjected the cell mix to only a membrane protein extraction buffer [33] or with the addition of a protease or phosphatase inhibitor such as PMSF [46,57]. Deng et al. and Wang et al. added Membrane Protein Extraction Reagent A containing PMSF [29,32,36,42,60,70]. Others only said that they had performed hypotonic lysis but didn’t describe any component of the buffer [24,41,43,49,65,66,75,89,97,104]. These buffers are shown in Table 2.

### 4.2. Homogenization

More than half of the articles employing hypotonic lysis treatment incorporated homogenization to optimize the extraction of membrane fragments [12,23,24,25,26,27,30,31,33,34,35,38,41,42,43,45,48,51,56,58,60,64,65,72,79,92,93,95,101,102]. In most of these studies [12,23,24,25,26,33,34,38,41,42,43,45,51,56,58,60,64,65,72,92,93,95,101,102] the common approach involved introducing lysed membrane fragments into a Dounce homogenizer. The fragments then underwent several passes or mechanical disruptions. The number of passes varied across experiments, ranging from 20 to 100. Notably, Kroll et al., Park et al., Jiang et al., and Wang et al. used a different system. They homogenized using a Polytron homogenizer for 15 [27,30,31,35] or 20 passes [79]. Jiang et al. homogenized instead the cells three times with an IKA T10 basic homogenizer [48].

### 4.3. Freeze-Thaw

While not a widely adopted strategy for this purpose, freeze-thaw has been employed in certain experiments [28,37,40,53,59,78,86,93,94,105,106]. This technique involves subjecting the cell suspension to multiple cycles of freezing and thawing, with the addition of only a phosphatase inhibitor to the suspension. In some cases, it has been utilized in combination with hypotonic lysis, submitting the lysed cells to several cycles of freezing in liquid nitrogen or at -80 °C and subsequent thawing at 37 °C [21,32,57,63,70]. Yao et al. performed a freeze-thaw treatment followed by sonication, without any previous hypotonic lysis [107].

### 4.4. Sonication

To harvest cell membrane fragments, a sonication treatment can be employed, which may involve the use of a bath sonicator [21,39,54,68,80,86,87,107] or ultrasonication with an ultrasonication device [22,36,44,50,52,69,73,90,99]. Soprano et al. utilized this method following hypotonic lysis and freeze-thaw treatments, placing the cells in a bath sonicator for 5 min [21]. Zhou et al. and Zhang et al. sonicated samples in a bath sonicator for 10 min after a hypotonic lysis treatment [80,87]. Nie et al. and Gan et al. applied repeated sonication steps in an ice bath [6,54]. Li et al. applied sonication after hypotonic lysis, subjecting the cells to 10 cycles of 3 s of ultrasonication at 150 W [22], while others homogenized the cell suspension using an ultrasonic disruptor [36,44,50,69,90]. Ultrasonication of the lysed membranes was used by several authors [52,73,99]. Dehaini et al. sonicated the cell suspension after a freeze-thaw treatment in a bath sonicator at 42 kHz and 100 W [86].

### 4.5. Other Methods

Another method employed for obtaining cell membrane fragments, either used alone or in combination with other techniques, is extrusion. In the studies by Chen et al. and Liu et al. extrusion is applied in conjunction with hypotonic lysis, occurring after the lysis process and before centrifugation to remove other cell components [95,104].

### 4.6. Summary

Upon reviewing all of the compiled articles, hypotonic lysis coupled with homogenization stands out as the overwhelmingly predominant method employed for membrane fragmentation in cells designated for coating. This approach has been consistently applied across a diverse range of cell types, encompassing both cancer and normal cells, and is independent of the specific cell type under investigation.

The hypotonic lysis buffer composed of 20 mM Tris-HCl pH 7.5, 10 mM KCl, 2 mM MgCl_2_, and 1 EDTA-free mini protease inhibitor tablet per 10 mL of solution, emerged as the most prevalent lysis buffer. Remarkably, this buffer was applied across various cell types, including melanoma, myeloma, triple-negative breast cancer, leukemia, and macrophages. Numerous other studies adopted similar lysis buffers based on Tris-HCl, either in combination with other compounds or inhibitors. Nevertheless, buffers incorporating Tris-HCl predominated, demonstrating their widespread usage and satisfactory results. In contrast, homogenization was predominantly carried out using a Dounce homogenizer, underscoring the effectiveness of this device in the membrane fragmentation process. The less commonly employed methods for membrane fragmentation were also applied to various cell types. Freeze-thaw was utilized for the fragmentation of macrophages, melanoma cells, erythrocytes, and platelets, while sonication was applied to cervical cancer, erythrocytes, and macrophages. These findings collectively suggest that there is no singular method universally valid for membrane fragmentation. Instead, there exist several reliable methods for this procedure, irrespective of the cell type chosen for coating. The selection of a specific method appears to be influenced by the availability of required materials in each laboratory. The advantages and disadvantages of each technique are detailed in Table 3.

## 5. Membrane Fragments Isolation

After the membranes have been fragmented, the next step involves recovering and isolating these fragments for their subsequent use in coating nanoparticles. Typically, the isolation stage includes 1 to 3 centrifugation steps to separate the remaining membrane materials. This process may be preceded or followed by a gradient separation to move other components away from the membrane fragments. Once the membrane fragments are obtained, they can undergo washing and/or lyophilization, or they may be directly resuspended if the nanoparticle coating process is scheduled immediately after isolation.

### 5.1. Centrifugation

To isolate membrane fragments from other cell components, most of the studies employed 1 to 3 cycles of centrifugation. Typically, a two-step process is followed. In the first centrifugation step, the mix undergoes a lower g force, approximately 3000× *g*, to precipitate the remaining cell components, and the supernatant was collected for the subsequent step. Some studies performed only this single centrifugation [28,29,37,46,48,74,76,81,82,84,94,106], whereas some others did a single centrifugation at higher g forces, such as 14,000× *g* [66], 15,000× *g* [90] or 21,000× *g* [107]. Others, seeking increased efficiency, resuspended the pellet, homogenized it, and subjected it to one or two additional centrifugations to recover more membrane fragments [12,23,25,34,38,45,56,58,62,73,90,92]. Numerous studies conducted the first centrifugation at 7000× *g* [63], 10,000× *g* [24,27,30,31,35,38,50,65], 16,000× *g* [87] or 20,000× *g* [41,72,101]. The second step involved one or two extra centrifugations of the supernatants from the first step to precipitate the membranes. This second step involved centrifugations at 3000× *g* to 8000× *g* [51,56,71,77] 10,000× *g* to 20,000× *g* [6,23,26,29,33,34,36,39,40,42,43,44,45,46,52,53,57,58,60,61,62,63,67,69,70,73,75,89,98,99,100,102], 30,000× *g* to 40,000× *g* [12,22,25,48,92], 100,000× *g* [24,41,47,65,72,79,101] or 150,000× *g* [27,30,31,35,38,50]. A final centrifugation or ultracentrifugation of the previous supernatant at around 15,000× *g* [51,77], 30,000× *g* to 40,000× *g* [45,48,71,92], 70,000× *g* [69], 80,000× *g* [58,61,95], or around 100,000× *g* [6,12,23,25,34,44,56,62,67,98,99,102] was also carried out in some cases.

### 5.2. Gradient

Certain experiments incorporated a gradient to enhance the performance of membrane fragments between the first and second centrifugations. This gradient took the form of a discontinuous sucrose density gradient, with weight/volume ratios of 55%, 40%, and 30%. The interface between 40% and 30% was then collected [48,92,93].

### 5.3. Washing

After isolation, the membrane fragments were at times washed in a 0.5–2 mM EDTA solution [12,25,27,31,35,58,102], sometimes with the addition of 10 mM Tris-HCl (pH 7.5) [12,25,58,102]. Alternatively, some studies washed the fragments with 1× PBS [28,36,66,74], HEPES [23], or 0.25 M sucrose [52].

### 5.4. Other Methods

In two investigations, a lyophilization step was implemented following the centrifugations. Parodi et al. lyophilized the isolated membranes before rehydrating them and storing them at 4 °C [92]. On the other hand, Bai et al. and Nie et al. directly lyophilized the membranes and stored them at −80 °C for future use [6,40,57,99].

### 5.5. Summary

In the isolation of membrane fragments, the predominant approach involved subjecting the fragments to one, two, or three centrifugation steps. Some experiments sought to enhance efficiency by incorporating additional steps such as resuspensions in lysis buffer and homogenizations, or by utilizing a sucrose gradient. However, the fundamental procedure typically comprised a combination of one to three centrifugation steps along with the washing of the isolated cell membrane fragments. The optimal number of centrifugations and the inclusion of a gradient appeared to be experiment-specific. While three-step centrifugation with additional lysis and homogenization steps might seem advantageous at first glance, it may not be universally necessary, and in some cases, omitting these extra steps could enhance efficiency. The decision on the specific approach likely depends on the unique requirements and outcomes of each experiment.

The different centrifugation steps and the g forces applied in each are dependent on which cell components are wanted and which ones need to be discarded. Centrifugation around 3000× *g* served to remove the nuclei and unbroken cells. Centrifugation steps at ca. 10,000× *g* or 20,000× *g* are used to remove mitochondria and other organelles. Finally, ultracentrifugation steps are performed to obtain the isolated cell membrane fragments. If the procedure does not require the elimination of organelles, the ultracentrifugation step can be omitted.

## 6. Nanoparticle Cores

Various types of nanoparticles were employed for coating, as shown in Table 4. Poly(lactic-co-glycolic acid) (PLGA) was overwhelmingly the most common choice in several studies [12,19,22,27,28,30,31,35,38,40,43,45,47,62,63,65,69,74,79,80,82,84,85,86,87,100,101,102,103,104,107]. However, the variety of nanoparticulate cores employed for membrane coating is extensive (Table 4).

Among the nanoparticles mentioned, hollow gold, hollow copper sulfide, melanin, and Fe_3_O_4_ nanoparticles, as well as NaYF_4_:Yb,Er core nanoparticles, serve a specific function beyond being carriers for cargo. The former are employed in photothermal therapy, where they are heated with light to generate hyperthermia, effectively killing the cancer cells targeted with the membrane coating [48]. On the other hand, the latter are utilized for photodynamic therapy, generating reactive oxygen species (ROS) when exposed to light [95]. MPBzyme ischemic stroke therapy, CoFc Ros production (Fenton reaction) to kill the tumor.

### 6.1. Cargoes Loaded into the Particles

In certain cases, the coated nanoparticles did not carry any additional load, as the nanoparticle itself was responsible for the desired therapeutic effect. For instance, in the study conducted by Jiang et al., melanin nanoparticles were employed for photothermal therapy without the need for an additional payload [49]. In the majority of other cases, nanoparticles were loaded with diverse substances tailored to the specific objectives of each research, as detailed in Table 5. These objectives ranged from chemotherapy to inhibiting molecular pathways, silencing genes, immune adjuvation (Figure 3a) or photosensitizing. The loaded substances included dexamethasone [22,35,47,89], doxorubicin [6,24,29,33,34,40,46,64,68,78,81,83,85,91,92,106], paclitaxel [62,74,76,94], cisplatin (Pt) [58], docetaxel [90], dacarbazine [55], SN-38 (primary active derivative of the pivot-al chemotherapeutic agent CPT-11, with enhanced efficacy in colorectal cancer) [98], methyl-triazeno-imidazole-carboxamide (MTIC) [66], KLA peptide (KLAKLAKKLAKLAK) [104], temozolomide [63,65], epirubicin [50], bortezomib (Figure 3b) [25], carfilzomib (CFZ) [102], ABT-737 [45], rapamycin [100], TPI-1 [33], mefuparib hydrochloride [6], hydroxychloroquine [60], NLG919 [53], aPD-1 [61], MLN4924 [43], R837 [28,67], L-γ-glutamyl-p-nitroanilide (GPNA) [52], bexarotene [72], siCdk4 [57], siRNASur [105], Ca^2+^ tar-geting siRNAs [24], mRNA transcripts for EGFP and CLuc [31], L-7, a TLR7 agonist [26], CpG oligodeoxynucleotide 1826 (CpG) [27], tetrakis(4-carboxyphenyl)porphyrin (TCPP), indocyanine green (ICG) [53,83,107], glucose oxidase [50], hemin [50], calcitriol [90], cannabidiol [101], Elamipretide [107], hySF (secreted factors from hypoxic adipose derived mesenchymal stem cells) [87], bone morphogenetic protein-2 (BMP-2) [75], minocycline hydrochloride (Mino) [36], low-molecular-weight fucoidan (LMWF) [103], bisphosphonate [56], Ag_2_S nanodots [37], AgAuSe quantum dots [72], uricase [96], recombinant human hyaluronidase, PH20 (rHuPH20) [80], 1,1′-dioctadecyl-3,3,3′3′-tetramethylindocarbocyanine perchlorate (DiI) [32,39,106], 1,1′-dioctadecyl-3,3,3′,3′-tetramethylindodicarbocyanine,4-chlorobenzenesulfonate salt (DiD) [19,30,82], 1,1′-dioctadecyl-3,3,3′,3′-tetramethylindotricarbocyanine iodide (DiR) [38], 3,3′-dioctadecyloxacarbocyanine perchlorate (DiO) [32,38] and IR780 [42].

### 6.2. Summary

PLGA has emerged as the overwhelmingly preferred material for nanoparticles, primarily due to its notable biocompatibility [22], biodegradability [45] versatile loading capabilities with various cargoes [12]. These characteristics make PLGA one of the most suitable materials for nanoparticle coating. The stabilization induced by the coating itself further enhances its utility since both cell membrane fragments and PLGA alone exhibit instability in physiological conditions. However, when united as a coated nanoparticle, their amalgamation remains stable until reaching the target cell. Upon reaching the target cell, the nanoparticle can be released to deliver the cargo effectively [12].

The selection of alternative nanoparticles might hinge on the therapeutic goal. For instance, if phototherapy or radiotherapy is desired, melanin nanoparticles, hollow gold, or copper sulfide nanoparticles may be better suited for the task. The choice of cargo for nanoparticles is entirely dependent on the therapeutic goal. Doxorubicin and dexamethasone stand out as the most frequently employed cargoes, owing to their well-established roles in cancer treatments, leveraging their chemotherapeutic [34] and anti-inflammatory [35] capabilities, respectively.

## 7. Membrane Coating of Nanoparticles

The coating of nanoparticle cores with isolated membrane fragments can be achieved through various methods, with extrusion and sonication being the most common. However, other techniques have also been employed, including a combination of sonication and extrusion. Additionally, in some studies, membrane vesicles were formed before being added to nanoparticles. The ratio of membrane to nanoparticles varies in each experiment, depending on the preceding steps and specific goals of the study. The general protocol for hybrid membrane-coated nanoparticles only derives from the standard on when the coating is applied, as both membrane fragments are mixed (Figure 4).

### 7.1. Coating after Vesicle Formation

Vesicle formation is achieved employing different methods that include extrusion and/or sonication [12,19,21,23,25,26,33,34,43,44,45,46,48,51,58,66,67,74,77,82,88,91,93,106]. Nanoparticle coating with vesicles involved similar methods to those described for membrane fragments, including extrusion [12,23,25,26,34,44,45,48,51,52,58,62,66,67,77], sonication [82,103], and a combination of sonication and extrusion [19,21,33,43,46,74,88,91,93,106]. In these cases, the sonication process typically involved using a bath sonicator for 2 [82], 5 [19,74,88,93,106] or 10 [21,46] minutes or ultrasonicated for 3 [91], 5 [103] or 15 min [43], whereas the extrusion methods entailed passing the fragments through 200 nm [66] or 400 nm polycarbonate porous membranes [12,21,23,25,26,33,43,44,45,46,51,58,62,67,77,88], or sequentially through 400 and 200 nm membranes [48,52,91,93,106] or 400 and 100 nm [19]. The coated nanoparticles were produced in a manner consistent with the rest of the process once these vesicles were formed.

### 7.2. Sonication

Sonication proved to be nearly as prevalent as extrusion, featuring independently in almost half of the procedures [23,24,27,29,30,31,32,33,35,37,38,40,47,50,52,54,57,59,68,71,72,78,79,82,85,86,89,99,101,102,104,107] and in combination with extrusion in some others [21,22,39,43,48,60,61,70,74,81,83,87,88,91,94,98,103]. The coating method typically involved sonication of the mixture of nanoparticles and membrane fragments in a bath sonicator for varying durations, of 2 [27,30,31,40,47,59,72,79,82,85,86,98,101,104], 3 [35,38], 5 [37,94,106], 6 [78,81], 10 [29,32,39,87,89,107], 20 [54] or 30 min [60,68,83]. In other studies, an ultrasonicator was utilized, sonicating the mixture in various intervals of a few seconds [33,50,57,102] or in a single treatment for 150 s [61] or 3 min [24].

Sonication was also employed in cases where membrane vesicles had been previously generated. In these instances, vesicles were sonicated along with nanoparticles in a bath sonicator for 30 s [88], or 2 [82,103], 3 [74], 5 [74], 10 [52] or 40 min [43], at a frequency of 53 kHz and a power of 100 W, or an amplitude of 50%.

### 7.3. Extrusion

Extrusion emerged as the predominant method for coating nanoparticles with isolated cell membrane fragments. This method is used in more than half of the investigations reviewed [6,12,19,23,25,26,34,36,41,42,44,45,46,49,51,53,58,62,63,64,65,66,67,69,70,73,76,77,80,84,90,93,94,96,97,100,105,106]. Additionally, in some other studies, extrusion was combined with sonication [21,22,39,43,48,60,61,70,74,81,83,87,88,91,94,98,103]. The coating procedure typically in-volved the coextrusion of both nanoparticles and membrane fragments, either in their fragmented state or having been previously transformed into vesicles. This coextrusion was performed for several cycles through a 100 nm [41,74,80,84], 200 nm [6,42,46,49,53,60,63,65,70,80,83,97,98], 220 nm [90], 400 nm [87], 800 nm [39] or 2 μm [73] polycarbonate membrane, or sequentially through polycarbonate membranes with pore sizes of 1000, 400, and 200 nm [64,105], 800, 400 and 200 nm [61], 400 and 200 nm [22,25,69,100], or 400 and 100 nm [94].

Extrusion was also employed in cases where membrane the vesicles had been previ-ously generated. In these instances, vesicles were coextruded with nanoparticles through 100 nm [19], 200 nm [12,23,26,44,51,58,66,67,93,106], 400 nm [34,43,45,62,88,103] or 800 nm [21] polycarbonate membranes or sequentially through 1000, 400, and 200 nm [48] or 200 and 100 nm [91] polycarbonate porous membranes.

### 7.4. Sonication-Extrusion

Several other studies employed a combination of both systems, involving a sonication treatment before implementing a standard extrusion procedure [21,22,39,43,48,60,61,70,74,81,83,87,88,91,94,98,103]. Two of those procedures performed the extrusion stage preceding the sonication of the mix [43,103].

### 7.5. Summary

Regarding nanoparticle coating, both sonication and extrusion appear to be valid methods. The frequency with which each method is employed suggests that they yield comparable results. However, a combination of both techniques could potentially enhance efficiency by combining the advantages of each. The advantages and disadvantages of these methods are shown in Table 6.

## 8. Discussion

An interesting factor to analyze after reviewing the methods is the membrane isolation efficiency, but almost none of the researchers gave information about it. Zou et al. mentioned how easy the erythrocytes were to isolate [77], while Fang et al. stated that their membrane isolation was successful [12]. Only Ferreira et al. gave specific results of the membrane isolation efficiency, reporting that 80% of the membrane was retained after isolation [65].

The coating efficiency is also an important factor to analyze since it shows how successful the coating was. In this regard, most of the researchers report a successful coating, showing the complete coating of the particles with TEM imaging or the analysis of zeta potential comparing the potential of the coated nanoparticles with those of the nude nanoparticle and the isolated membrane. Only 3 of the reviewed articles gave an exact value of coating efficiency. Liu et al. reported a 90.21% efficiency [104], which is in line with the reports of complete or almost complete coating given by all of the investigations that analyzed it with TEM and zeta potential. Conversely, Li et al. report a 21% coating efficiency with a sonication method [50], and Liu et al. measured the coating with a fluorescence quenching essay where they used a quencher that cannot cross membranes and therefore only affects their uncoated parts, state that up to 90% of the nanoparticles are only partially coated and 60% of them are only 20% coated [23]. These results open the door for future improvements to the coating techniques.

Most of the coatings caused an increment of around 10 to 30 nm to the diameter of the nanoparticles. But there were many cases where the increase was notably higher, such as Liu et al. (66 nm) [99], Ren et al. (59 nm) [52], Li et al. (56 nm) [103], Bu et al. (80 nm) [73], or Li et al. (140 nm) [53]. These results can be attributed to an imperfect membrane coating of the nanocarriers, either by having more than one layer of membrane fragments or by the creation of aggregates of those fragments on the surface of the particle. Conversely, Huang et al. report a more exceptional result where they observed a reduction of the size of the nanoparticles, diminishing from 150.1 to 137.3 nm [66]. The researches attribute this decrease in size to the pressures to which the particles are subjected to during the extrusion process [66].

The particle-membrane interactions were covered by only a handful of the reviewed articles, since most of them were focused on the effects of the cargo loaded on the nanoparticles on the cells. Despite that, some articles give interesting information about these interactions. Ferreira et al. and Scully et al. explain that the coating is achieved by electrostatic interactions that favor the right-side orientation of the membrane [45,65]. Chen et al. and Liu et al. also state that negatively charged nanoparticles give better results than positively charged nanocarriers due to their electrostatic interactions [5,23]. Luk et al. stated that the negatively charged cores created a more subtle interaction, allowing the membranes to retain their structure and fluidity, whereas the positively charged cores created strong electrostatic interactions that can cause the collapse of the membrane and thus create aggregates of nanoparticles and membrane fragments [109]. Mornet et al. went further and analyzed the effect of differently charged membranes on the coating. They observed that highly negative membranes didn’t achieve a successful coating, but moderately negatively charged membranes were able to completely coat the nanoparticles [110]. Xia et al. attribute these interactions to the presence of dense negatively charged sialic acid moiety present in the outer membrane side, that allows the right side of the membrane to coat the nanoparticles when a negatively charged core is used but causes the formation of aggregates when positively charged nanoparticles are used due to these negative charges located in the outer side of the membrane [111]. Zhao et al. and Zhang et al. state that a higher concentration of H^+^ in the tumoral microenvironment favors the dissociation of the membrane and the nanoparticle, allowing for a faster release of the cargo [24,94].

The biological and micro/nano interactions responsible for tissue-specific therapeutics using these nanoparticles are very diverse. The most common approach was to take profit from the homotypic targeting allowed by the “self-recognition” molecules present on the target tissue [45], especially among those who wanted to target cancers with patient-derived tumor cells, since cancer cells have surface antigens that allow multicellular aggregation through homophilic adhesion domains [100]. Some of them rely on the presence of proteins in the membrane coat of the nanocarriers that attach to receptors of the target cells, allowing thus their internalization via endocytosis, such as Tiwari et al. [55], who relied on the presence of heparanase, syndecan-1 and glypican-1, that target HSPG receptors, unchaining the endocytosis. The particles that were designed to avoid immune recognition profited from immune and other blood cells components, especially from macrophages and erythrocytes, respectively, such as macrophages’ SIRPα receptor, to which the CD47 proteins of the membranes of the donor cell bind to be recognized by the macrophages and avoid phagocytosis [112]. Some opted for the decoration of membranes with targeting molecules, such as aptamers, that target the tumors [78]. Another alternative was to genetically modify the donor cells to overexpress a protein that targets a specific protein from the target tissue, such as the rabies viral glycoprotein used by et al. to target acetylcholine receptors on cerebrovascular endothelial cells and nerve cells [72]. Another example of this is the use of antibodies linked to the membrane, designed to target the aimed cells [39].

The release kinetics were given by almost all of the reviewed articles, but most of them only studied the difference of released cargo at different pH values. As expected, more cargo was released and also in a faster way when the coated nanocarriers were in more acidic conditions, such as those present at the tumor microenvironments, than in normal physiological conditions (i.e., pH 7.4) [55,78]. But among those who actually compared coated and non-coated particles, there were different results. Some researchers such as Qi et al., Zhang et al., Li et al., and Lin et al. report similar release kinetics between both types of carriers, with a minimal difference in speed and total release, as coated nanoparticles were a bit slower and released a bit less cargo than their non-coated counterparts [22,87,91,98]. Conversely, Ma et al. observed that coated nanoparticles released less cargo at pH 7.4 but at pH 5.5 were more effective in the release than the non-coated ones [100]. Others, such as Li et al. and Chen et al. observed that coated nanoparticles released 10% less of the total cargo than those that were not coated during the first 12–24 h, but in the long term (5–7 days) both end up releasing the same amount of cargo [60,62]. Tian et al. observed a great difference in released cargo between coated and non-coated nanoparticles (16.85% against 40.1%), releasing thus less cargo during circulation and improving drug delivery [74]. A similar result is reported by Qu et al., who observed a similar difference but both coated and non-coated nanoparticles release higher amounts of cargo (33% versus 50%) [25], and by Scully et al., who reported a 12% release of cargo after 24 h and 16% after 48 h in coated nanocarriers, whereas the non-coated released 30 and 37%, respectively [45]. Parodi et al. studied the release kinetics of two different cargos (doxorubicin and BSA) [92]. There were very significant differences in the release of both cargos between coated and non-coated carriers, being 20% against 45% release of doxorubicin after 3 h, and 15 versus 25% after 3 h and 80 versus 90% after 48 h, respectively [92]. In Liu et al.’s study, non-coated cores were able to deliver the whole cargo after 72 h, but their coated counterparts only released 50% of it in those 72 h, requiring 120 h to release 90% of the cargo [101]. Liu stated that the use of PEG and the membrane coating improved the stabilization of the nanoparticles, allowing the reported better retention of the cargo in the nanocarriers [101]. Du et al. saw almost no difference in release between coated and non-coated nanocarriers at pH 7.4 (both around 11%) but noticed a significant 16% difference at pH 5.0 [64]. Despite not releasing less at physiological conditions and being less efficient at tumor conditions, the low release at pH 7.4 allows for an enhanced cargo accumulation at tumor sites and a reduction of toxicity to other tissues [64]. Xie et al. noted that at pH 7.4 coated carriers released much less cargo than the non-coated ones (24.3% against 37.9%), but at pH 5.5, both released more similar amounts (76.1% versus 84.1%) [76]. Gong et al. reported a bigger difference at pH 7.4 (40% against 65%), but at pHs 5.5 and 4.7 those differences are reduced significantly, especially at pH 4.7, where the difference is almost negligible [40]. These results from Xie et al. and Gong et al. show that the coating protects the nanoparticles and avoids the loss of cargo before arriving at the tumor, improving thus the loading capacity and the drug release behavior [76].

These coating techniques were evaluated through a comparison between cell membrane-coated nanoparticles and their non-coated counterparts and/or free cargoes. In all studies conducting cellular uptake analyses, improvements were consistently observed compared to non-coated nanocarriers and free substances. While some studies reported a twofold increase in uptake, others, such as Fang et al., noted a remarkable 40-fold improvement [12]. Certain investigations extended their analysis by comparing uptake in the target cell type with other cell types to assess specificity. For instance, Bai et al. observed significantly higher uptake in the target cells compared to other cell types [57]. Furthermore, certain studies prioritized investigating immune avoidance, noting a reduction in phagocytosis of coated nanoparticles by macrophages compared to non-coated nanocarriers [38,82,92,101]. In summary, cell membrane-coated nanoparticles consistently demonstrated improvements in uptake, specificity, or immune evasion compared to their non-coated counterparts.

## 9. Future Directions

While this review primarily concentrates on the cell membrane coating of nanoparticles designed for combating cancers, the application of biomimicry extends beyond oncology. This promising technique has found utility in diverse areas such as gene editing, including the induction of gene expression [31], gene silencing [57], detoxification [79,82], ischemic stroke therapy [101], immune modulation [113], and antibacterial vaccination [36,114,115]. The versatility of biomimicry underscores its potential across various fields of research and therapeutic applications.

Further research is needed to enhance the effectiveness of cell membrane-coated nanoparticles, as well as to improve their coating efficiency. While they are already more effective than naked nanoparticles, improvements in targeting ability and residence time are areas of focus. Several novel methods have been explored for this purpose, including modified lipid insertion, membrane hybridization, and genetic modification of source cells [116,117]. Modified lipid insertion involves introducing modified lipids into the coated nanoparticles to enhance their fusion and ligand binding properties. For instance, modified lipid insertion has been shown to improve fusion properties [21] and ligand binding properties [84]. Membrane hybridization combines characteristics from different source cell membrane fragments [86]. For example, creating an erythrocyte-melanoma hybrid coat provides the coated nanoparticles with both the prolonged circulation time of erythrocytes and the homotypic targeting capabilities of melanoma cell membranes [29,46]. Genetic modification of source cells involves introducing specific membrane proteins or lipids not present in the original, non-modified cells. This genetic modification provides the coated nanoparticles with the ability to specifically target new ligands, thereby improving their targeting ability [30].

## Figures and Tables

**Figure 1 ijms-25-02071-f001:**
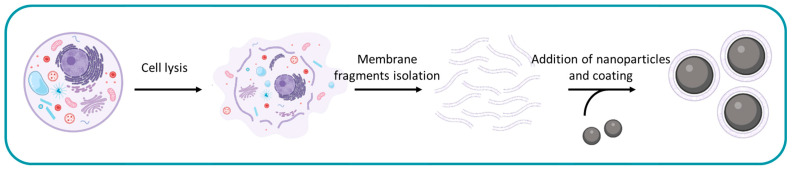
Three main steps for obtaining cell membrane-coated nanoparticles: cell lysis and membrane fragmentation, isolation of membrane fragments, and coating selected nanocarriers. The figure has been created with BioRender.com.

**Figure 2 ijms-25-02071-f002:**
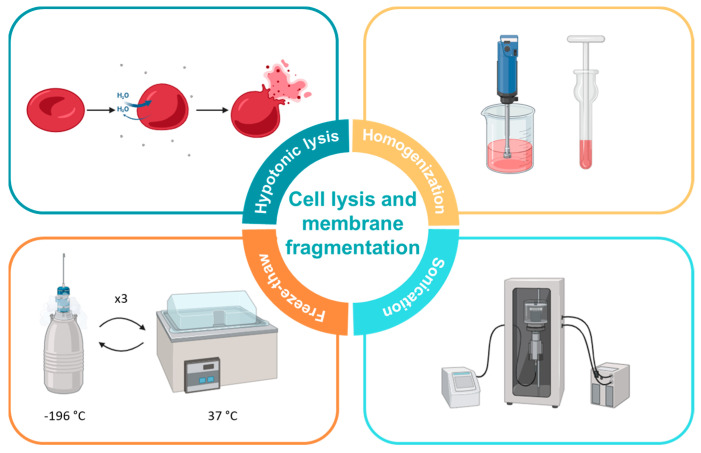
Main strategies used for cell membrane fragmentation: hypotonic lysis; homogenization with probe homogenizer or dounce homogenizer; freeze-thaw and sonication. The figure has been created with BioRender.com.

**Figure 3 ijms-25-02071-f003:**
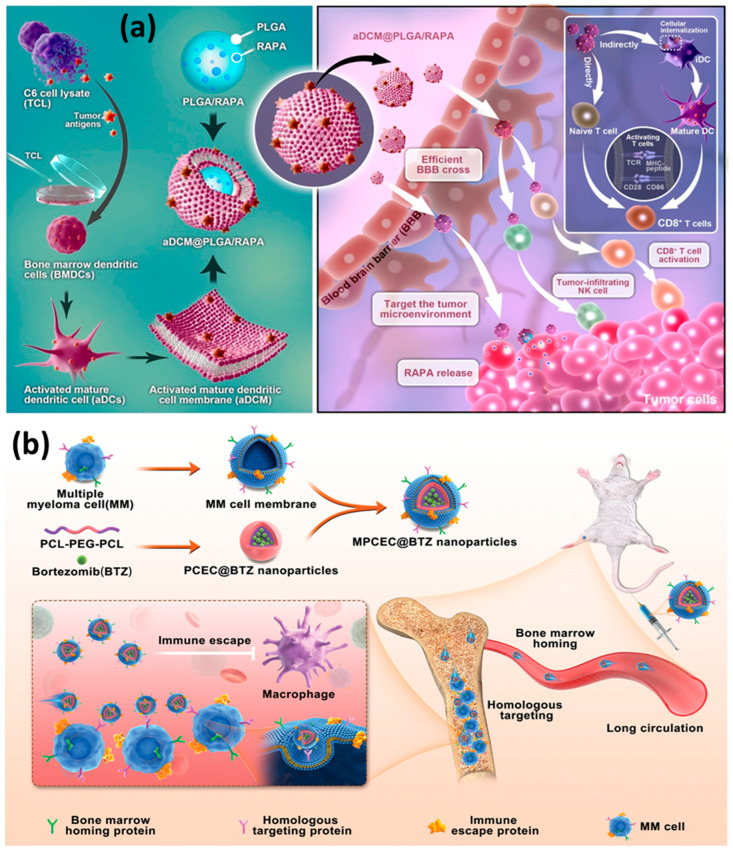
Examples of cell membrane-coated nanoparticles. (**a**) Sequential process of activated dendritic cells (aDCs) and the synergistic effect of activated mature dendritic cell membrane (aDCM)-coated nanoplatform, rapamycin (RAPA)-loaded poly(lactic-co-glycolic acid) (PLGA), named aDCM@PLGA/RAPA, drug delivery nanoplatform, directly or indirectly to activate immunotherapy. Adapted from Ref. [100]. Copyright© 2023 American Chemical Society. (**b**) Scheme of Multiple myeloma (MM)-cell-membrane-coated poly(ε-caprolactone)–poly(ethylene glycol)–poly(ε-caprolactone) (PTEC) nanoparticles for treatment of multiple myeloma. After intravenous injection, these biomimetic nanoparticles could enter the bone marrow (BM) cavity due to the bone marrow homing (BMH), then target the tumor cells through homologous targeting. Adapted from Ref. [25]. Wiley© 2012.

**Figure 4 ijms-25-02071-f004:**
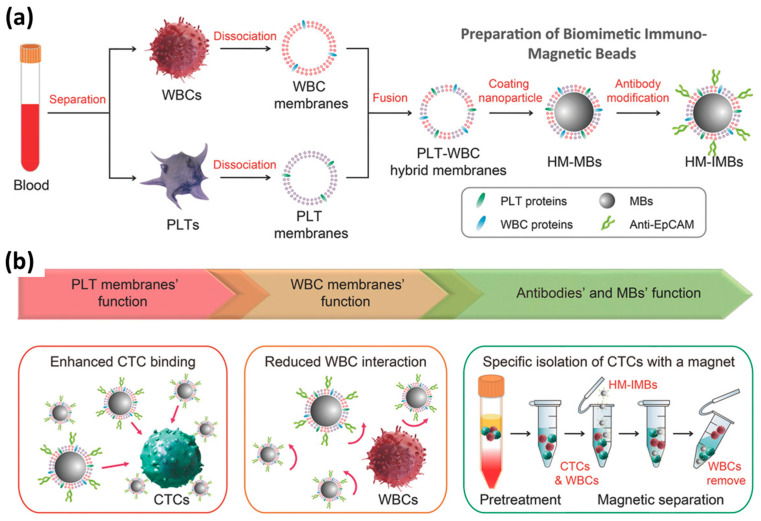
Scheme of the preparation of hybrid membrane-coated immunomagnetic beads (HM-IMBs) for high-performance isolation of circulating tumor cells (CTCs). (**a**) Platelet (PLT) and leukocyte (WBC) membranes, along with their associated proteins, were independently separated from blood samples, fused, and then coated onto MBs. Then, the resulting PLT–WBC HM-coated MBs were surface-modified with antibodies to form HM-coated immuno-MBs. (**b**) HM-IMBs inherited enhanced CTCs binding from PLTs and the property of reduced interaction with homologous WBCs from WBCs, was used for high-efficiency and high-purity isolation of CTCs. Copy from Ref. [93], Wiley^©^ 2018.

**Table 1 ijms-25-02071-t001:** Donor cell types for nanoparticle coating applications.

Donor Cell	Cell Lines	Application	References
Cervical and ovarian cancer	HeLa	Homologous targeting	[21,22,23,24]
Multiple myeloma	ARD, KMS11, 5TGM1		[25]
Melanoma	B16-F10, MDA-MB-435		[12,26,27,28,29,30,31,32]
Leukemia	CHRF-288-11, C1498, RAW264.7, THP-1, Jurkat, HL-60		[23,33,34,35,36,37,38,39,40,41,42,43,44]
Breast cancer	4T1, MCF-7, MDA-MB-231, MDA-MB-468		[6,37,40,45,46,47,48,49,50,51,52,53,54,55,56]
Colon carcinoma	CT-26		[23,57]
Head and neck squamous cell carcinoma	CAL 27, SCC7		[58,59,60,61]
Lung cancer	NCI-H460, A549		[54,62]
Glioma	GL261, C6, U87MG		[63,64]
Glioblastoma	U251		[65,66]
Prostate cancer	RM-1		[67]
Liver cancer	HepG2		[68]
Fibroblasts	NIH 3T3		[49,69]
Embryonic kidney cells	HEK293		[70]
Vaginal endothelial cells	VK2/E6E7		[71]
Neural stem cells	Primary cells		[72]
Microglia	HMC3		[66]
Keratinocytes	Hacat		[73]
Mesenchymal stem cells	Primary cells		[74,75,76,77,78]
Neuroblastoma	Neuro-2a	Neurotoxin capture	[79]
Erythrocytes	Primary cells	Cancer tissue targeting	[19,29,46,48,80,81,82,83,84,85,86,87,88,89,90,91]
Leukocytes	Primary cells	Avoidance of immune recognition	[89,92,93,94,95,96,97,98,99,100,101,102,103,104]
Platelets	Primary cells	Cancer cell binding ability	[48,86,87,93,105,106,107]

**Table 2 ijms-25-02071-t002:** Hypotonic lysis buffers used to obtain cell membrane fragments.

Lysis Buffer Used ^1^	References
Tris-HCl-based hypotonic buffers	[6,12,23,25,27,30,31,34,38,45,50,51,54,56,58,62,63,67,69,71,72,73,77,92,95,100,101,102]
PBS-based hypotonic buffers	[21,46,61,76,81,82,87,88,90,91]
HEPES-based hypotonic buffers	[22,48,93]
EGTA-based hypotonic buffers	[35,79]
NaHCO_3_-based buffers	[39,64]
Double distilled water	[103]
Unspecified hypotonic buffers	[24,26,32,33,36,41,42,46,49,57,60,65,66,75,86,89,97,104]

^1^ The buffers also carried protease inhibitors, and in some cases, phosphatase inhibitors.

**Table 3 ijms-25-02071-t003:** Advantages and disadvantages of the membrane fragmentation techniques.

Technique	Advantages	Disadvantages
Hypotonic lysis	Maintains membrane characteristics Compatible with downstream applications	Typically necessitates a combination with other techniques to obtain the fragments.
Homogenization	Maintains membrane characteristics	Typically necessitates a combination with other techniques to obtain the fragments
Freeze-thaw	Simplicity	Potential damage to temperature-sensitive membrane proteins Impact on the activity of sensitive enzymes Cryoconcentration
Sonication	Fastest method	Potential damage to temperature-sensitive membrane proteins Generation of free radicals

**Table 4 ijms-25-02071-t004:** Nanoparticles used for membrane coating.

Nanoparticles	Size Range (nm)	Function	References
PLGA	50–300	Drug loading	[12,19,22,27,28,30,31,35,38,40,43,45,47,62,63,65,69,74,79,80,82,84,85,86,87,100,101,102,103,104,107]
Polystyrene	100–200		[21]
PCEC	50–150		[25]
MPEG-PLGA	50–150		[26]
PCN-224	50–150		[57]
PEG-PLGA	25–150		[34,72]
PEGDA	100–150		[81]
Gelatin	50–100		[58]
Poly(β-amino ester)	–		[94]
ZIF-8 MOF	100–300		[50,96,105]
Spherical nonporous SiO_2_ nanoparticles	50–150		[23]
Mesoporous silica nanoparticles	150–200		[6]
Colloidal silica nanoparticles	200–250		[77]
Porous silica	150–200		[56]
Chitosan-silica nanoparticles	100–200		[24,68]
Nanoporous silica	–		[92]
Silk fibroin	100–150		[36]
Graphene oxide	150–200		[83]
Magnetic beads	50–150		[93]
Fe_3_O_4_@SiO_2_ nanoparticles	50–450		[37]
Heparan sulfate	100–200		[90]
PMBEOx-COOH	25–75		[67]
Curdlan	50–150		[91]
PFC	150–200		[71]
Pluronic F127 nanomicelles	50–250		[53]
Liposomes	100–150		[33,98]
CB[7]-PEG-Ce6 polymer	100–200		[66]
Polydopamine-fructose-curcumin nanoparticles	100–200		[99]
Hollow gold nanoparticles	100–200	Chemo/Photothermal therapy	[46,78]
Hollow copper sulfide nanoparticles	150–250		[29]
Polypyrrole	100–150		[106]
Melanin nanoparticles	200–250	Photothermal therapy	[48]
Fe_3_O_4_ nanoparticles	50–250		[39,59]
Hollow polydopamine	150–200		[32]
DHTDP	50–150		[51]
BiOI nanodots	5–10	Radiotherapy	[97]
NaYF_4_:Yb,Er nanoparticles	50–100	Photodynamic therapy	[95]
NaYF_4_:Nd_5_@NaYF_4_	100–200	Imaging	[49]
NaGdF_4_:Yb,Tm nanoparticles	100–150		[88]
Gd MOF	150–200		[61]
MPBzyme	100–200	Ischemic stroke therapy	[41]
Co-Fc MOF	250–300	ROS production	[60]
BTO nanoparticles	50–150		[70]
MnO_2_	25–150		[44,64,76]
IrO_2_	50–150		[52]
CuPt nanoalloys	25–50		[54]
Fucose-based CQDs	5–10		[55]
Gelatin microribbon scaffolds	200–300	Bone regeneration	[75]
AMPNP	50–100	Antibacterial function	[67]

**Table 5 ijms-25-02071-t005:** Cargoes loaded in the nanoparticles.

Load	Use/Function	Nanoparticles	Bioactive Loading	References
Dexamethasone	Anti-inflammatory drug Chemotherapy, radiotherapy and immunotherapy	PLGA	2–10% ^3^	[22,35,47]
		Hollow copper sulfide	45.52% ^2^	[89]
Doxorubicin	Chemotherapy	NPS	-	[92]
		HGNPs	31–37% ^3^	[46,78]
		PEG-PLGA	14.2 ± 2.4% ^1^	[34]
		PEGDA	15% ^3^	[81]
		GO	42.9% ^3^	[83]
		DCuS	87.7% ^1^	[29]
		PLGA	9–10% ^1^	[40,85]
		Mesosporous silica	-	[6]
		Liposome	40% ^3^	[33]
		Chitosan-silica	18–33% ^3^	[24,68]
		Polypyrrole	-	[106]
		MnO_2_	40–70% ^3^	[64]
		Curdlan	-	[91]
Paclitaxel		PLGA	4–16% ^2^	[62,74]
		Poly(β-amino ester)	9.88% ^3^	[94]
		MnO_2_	-	[76]
Cisplatin (Pt)		Gelatin nanoparticles	12.55% ^3^	[58]
Docetaxel		Heparan sulfate	9–10% ^2^	[90]
Dacarbazine		Fucose-based CQDs	-	[55]
SN-38		Liposomes	5.54 ± 0.73% ^1^	[98]
MTIC		(CB[7]-PEG-Ce6)	5.42% ^3^	[66]
KLA peptide	Induces apoptosis	PLGA	-	[104]
Temozolomide	Alkylating agent	PLGA	8% ^3^	[63]
Epirubicin	Immunogenic cell death inducer	ZIF-8	-	[50]
Bortezomib	Proteasome inhibitor	PCEC	2.87 ± 0.51% ^3^	[25]
Carfilzomib	Proteasome inhibitor	PLGA	3.74 ± 0.28% ^3^	[102]
ABT-737	Bcl-2 inhibitor	PLGA	5–10% ^1^	[45]
Rapamycin	Specific inhibitor of the mTOR signaling pathway [108]	PLGA	11.39% ^2^	[100]
TPI-1	Inhibitor of the downstream effector molecule SHP-1	Liposome	40% ^3^	[33]
Mefuparib hydrochloride	poly(ADP-ribose) polymerase inhibitor	Mesoporous silica	-	[6]
Hydroxychloroquine	Autophagy inhibitor	Co-Fc	12.81 ± 4.21% ^3^	[60]
NLG919	IDO-1 enzyme inhibitor	Pluronic F127	5.08% ^3^	[53]
aPD-1	PD-1 inhibitor	Gd-MOF	-	[61]
MLN4924	Neddylation inhibitor	PLGA	10% ^3^	[43]
R837	Antagonist against TLR-7	PLGA	8% ^1^	[28]
		PMBEOx-COOH	6.1% ^3^	[67]
L-γ-glutamyl-p-nitroanilide (GPNA)	Glutamine transporter antagonist (Glycolysis inhibition)	IrO_2_	-	[52]
Bexarotene	hydrophobic retinoid X receptor (RXR) antagonist	PEG-PLGA	43.24% ^3^	[72]
siCdk4	Knocks down Cdk4	PCN-224	1.3 μg/mg	[57]
siRNA^Sur^	Knocks down Survivin	ZIF-8	-	[105]
Ca^2+^ targeting siRNA	Knocks down the expression Ca^2+^ channels	Chitosan-silica	1.12% ^3^	[24]
mRNA transcripts for EGFP and CLuc	Silence EGGP and CLuc	PLGA	1 μg/mg	[31]
L-7	Immune adjuvant	MPEG-PLGA	2.69% ^3^	[26]
CpG oligodeoxynucleotide 1826	Immunological adjuvant that triggers the maturation of antigen-presenting cells	PLGA	1 nmol/mg	[27]
TCPP	Photosensitizer	MPEG-PLGA	4.84% ^3^	[26]
Indocyanine green (ICG)	Photothermal agent	Graphene oxide	10.7% ^3^	[83]
		Pluronic F127	10.26% ^3^	[53]
		PLGA	-	[107]
Glucose oxidase	Mediators of the cascade generation of ROS	ZIF-8	-	[50]
Hemin			-	
Calcitriol	Anti-metastasis agent	Heparan sulfate	2.92 ± 0.16% ^2^	[90]
Cannabidiol	Neuroprotective product	PLGA	3.9 ± 0.2% ^3^	[101]
Elamipretide	Antioxidant	PLGA	-	[107]
hySF	Vascular regeneration	PLGA	-	[87]
BMP-2	Boosting bone regeneration	Gelatin microribbon scaffolds	-	[75]
Minocycline hydrochloride	Antimicrobial agent	Silk fibroin	7.86% ^3^	[36]
LMWF	Anti methicillin-resistant *Staphylococcus aureus*	PLGA	4.7% ^1^	[103]
Biphosphonate	Chelator for ^89^Zr radiolabeling	Porous silicon	-	[56]
Ag_2_S nanodots	Biosensing and bioimaging	Fe_3_O_4_@SiO_2_ nanoparticles	-	[37]
AgAuSe quantum dots	Bioimaging	PEG-PLGA	10% ^3^	[72]
Uricase	PoC study	MOF	-	[96]
DiI	Fluorophore, PoC study	Hollow dopamine	-	[32]
		Fe_3_O_4_	-	[39]
		SiO_2_	-	[77]
DiD		PLGA	0.2% ^1^	[82]
DiR			0.1% ^1^	[38]
DiO			0.1% ^1^	
		Hollow polydopamine	-	[32]
IR780		AMPNP	-	[42]

^1^ Load weight/polymer weight. ^2^ Load weight/total nanoparticle weight. ^3^ Not specified.

**Table 6 ijms-25-02071-t006:** Advantages and disadvantages of membrane coating techniques.

Technique	Advantages	Disadvantages
Sonication	Allows the fusion of multiple cell membranes from different cell types Favors right-side out orientation of the membranes	Potential damage to temperature-sensitive membrane proteins Generation of free radicals
Extrusion	Allows the creation of multi-layer structures Does not denature proteins	Can cause a reduction in drug loading It is not applicable for irregularly shaped nanoparticles
Sonication-extrusion	Combines the advantages of both	Retains the disadvantages of both, except the inability to coat irregularly shaped nanoparticles

## Data Availability

Not applicable.

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
