# Peer review of "Cell Membrane-Coated Nanoparticles for Precision Medicine: A Comprehensive Review of Coating Techniques for Tissue-Specific Therapeutics"

_ijms, 2024, doi:10.3390/ijms25042071_

Round 1
Reviewer 1 Report
Comments and Suggestions for Authors
In this manuscript, the authors reviewed fabrication of cell membrane-coated nanoparticles, and also reported their applications in tissue-specific therapeutics. This work seems to be useful in this field. However, major revisions are needed to address the following problems before further consideration of publication:
1. Composite figures should be added (one-three) in the related sections to show typical research results contained in the review.
2. The Introduction is suggested to come straight to the point, and references should be enriched. The related advances and applications of biosynthesis and also precise therapy should be added including: 10.1002/sstr.202200356, 10.3390/molecules28083299.
3. This review is mainly focused on two parts: fabrication and applications. For a comprehensive review, the authors are suggested to improve the manuscript structure with clear organized sections. A brief summary of applications and the corresponding performances can be contained.
4. Section 5 and 7 are both important. Detailed explanation with recent advances and development should be added since the contents seem to be simple and brief.
5. The depth of presentation could be improved if the authors provided some insights of biological and micro/nano interactions responsible for tissue-specific therapeutics using these nanoparticles.
6. The perspectives or challenges for future should be discussed in detail, where a schematic illustration can also be designed.
7. All the figures need to be created in consistent layout to improve the readability. The current styles seem to be disordered and not normative. Besides, the contents and captions such as Figure 1-2 are quite simple and should be explained in detail.
8. The format of references should be check thoroughly considering some citing errors.
Author Response
Manuscript Number: ijms-2785833
REVIEWER #1
In this manuscript, the authors reviewed fabrication of cell membrane-coated nanoparticles, and also reported their applications in tissue-specific therapeutics. This work seems to be useful in this field.
We express our gratitude to the reviewer for her/his positive comments and for finding our work interesting.
However, major revisions are needed to address the following problems before further consideration of publication:
- Composite figures should be added (one-three) in the related sections to show typical research results contained in the review.
Reply to referee.
We appreciate the comment, and more composite figures have been added. Figure 3 depicts two high-impact papers about cell membrane-coated nanoparticles, one from dendritic cells (Figure 3a) and the other from multiple myeloma cells (Figure 3b). Additionally, Figure 4 illustrates the scheme for the top-level publication detailing the preparation of hybrid membrane-coated immunomagnetic beads for high-performance isolation of circulating tumor cells.
- The Introduction is suggested to come straight to the point, and references should be enriched. The related advances and applications of biosynthesis and also precise therapy should be added including: 10.1002/sstr.202200356, 10.3390/molecules28083299.
Reply to referee.
We appreciate the reviewer's insightful feedback regarding the need for improvement in the references. In response, we have made substantial revisions to include a more extensive list of references in this updated version.
However, upon careful consideration, we have observed that the two references recommended by the reviewer do not seem directly relevant to the content of our paper. The first one is based on the biologically driven micro/nanofabrication, assembly, and actuation based on microorganisms. The second one talks about the Green-synthesized zero-valent iron nanoparticles (ZVI-NPs) using Nigella sativa seed extract for heavy metal remediation
- This review is mainly focused on two parts: fabrication and applications. For a comprehensive review, the authors are suggested to improve the manuscript structure with clear organized sections. A brief summary of applications and the corresponding performances can be contained.
Reply to referee.
We agree with the referee's opinion and the paper has been greatly reorganized (all changes are highlighted in yellow). Besides, the applications are already summarized in the tables. The specific results of each application have not been commented due to space constraints and this is note the main objective of the paper.
- Section 5 and 7 are both important. Detailed explanation with recent advances and development should be added since the contents seem to be simple and brief.
Reply to referee.
We have taken the suggestions into consideration and have incorporated new references, including information from 2023 and even 2024. Furthermore, we have expanded sections 5 and 7 with detailed explanations of recent advances and developments, addressing the perceived simplicity and brevity of the contents.
- The depth of presentation could be improved if the authors provided some insights of biological and micro/nano interactions responsible for tissue-specific therapeutics using these nanoparticles.
Reply to referee.
Following the reviewer's recommendations, this part has been included at the end of the discussion (changes are highlighted in yellow).
- The perspectives or challenges for future should be discussed in detail, where a schematic illustration can also be designed.
Reply to referee.
A new paragraph has been added about perspectives and challenges
- All the figures need to be created in consistent layout to improve the readability. The current styles seem to be disordered and not normative. Besides, the contents and captions such as Figure 1-2 are quite simple and should be explained in detail.
Reply to referee.
We thank the referee for this comment. The figure captions have been enlarged and detailed for better understanding
- The format of references should becheck thoroughly considering some citing errors.
Reply to referee.
We thank the referee for this comment. We have revised and enhanced all the references
Reviewer 2 Report
Comments and Suggestions for Authors
The work “Cell Membrane-Coated Nanoparticles for Precision Medicine: A Comprehensive Review of Coating Techniques for Tissue-Specific Therapeutics” tells a detailed story about how to coat cell membrane to nanoparticles, the statement is logical and reasonable, and it stated a basic problem to many corresponding researchers, which will attract plenty of readers. The figures authors drew are also nice and scientific, the references are relatively new.
In conclusion, it is a good work that deserves to be published. However, before that, the following questions and suggestions should be considered and solved first.
Major:
1. In part 4, a comparation of different strategies used for cell membrane fragmentation (hypotonic lysis; homogenization with probe homogenizer or dounce homogenize; freeze-thaw and sonication) should be written as a paragraph or it’s better to construct a table to list the advantages and disadvantages of the methods. The same as in part 5, membrane fragments isolation methods (centrifugation (once, twice, and third times, low speed first, and high speed first) and gradient) should also be compared and listed as a table to make reader and researcher be clear which method is suitable for them. In part seven, the comparation among “extrusion, sonication, sonication&extrusion” should also be stated.
2. Part 10 (conclusions) can be deleted since the contents are repeated with part 8 and 9.
Minor:
1. In the title, the statement of “for tissue specific therapeutics” can be deleted since the paper mainly focuses on the coating technique.
2. In introduction part, I’d suggest citing the following 3 fresh and comprehensive papers to instead ref 1-9 efficiently, since they include all the advantages of nanomedicines and nanoencapsulations the authors mentioned:
https://www.sciencedirect.com/science/article/pii/S1748013218306388?via%3Dihub
https://www.sciencedirect.com/science/article/pii/S0378517318300784?via%3Dihub
https://pubs.acs.org/doi/full/10.1021/acsanm.3c04487
3. What’s the unique advantage of membrane-coated NPs compared to other similar components such as EV/lipid-coated NPs? As far as I know, there are some reports that EV can also target cells. It’s better to state this in the introduction part of manuscript.
4. It’s better to add the exact cell line in Table 1 as a new row.
5. In Table 4, the corresponding NPs should also be listed as a row or combine it with Table 3 together.
6. In Figure 3, I’d suggest selecting more TOC/Scheme figures from other references you referred to (line 271-283) to replace Figure 3b-3e to show more interesting works to readers. Since you also didn’t talk in detail about these figures (3b-3e) in the text.
7. In part 9, I believe another direction is to increase the efficiency of membrane isolation and coating rate for NPs since both are too low at present.
Author Response
Manuscript Number: ijms-2785833
REVIEWER #2
The work “Cell Membrane-Coated Nanoparticles for Precision Medicine: A Comprehensive Review of Coating Techniques for Tissue-Specific Therapeutics” tells a detailed story about how to coat cell membrane to nanoparticles, the statement is logical and reasonable, and it stated a basic problem to many corresponding researchers, which will attract plenty of readers. The figures authors drew are also nice and scientific, the references are relatively new.
In conclusion, it is a good work that deserves to be published. However, before that, the following questions and suggestions should be considered and solved first.
Major:
- In part 4, a comparation of different strategies used for cell membrane fragmentation (hypotonic lysis; homogenization with probe homogenizer or dounce homogenize; freeze-thaw and sonication) should be written as a paragraph or it’s better to construct a tableto list the advantages and disadvantages of the methods. The same as in part 5, membrane fragments isolation methods (centrifugation (once, twice, and third times, low speed first, and high speed first) and gradient) should also be compared and listed as a table to make reader and researcher be clear which method is suitable for them. In part seven, the comparation among “extrusion, sonication, sonication&extrusion” should also be stated.
Reply to referee.
We thank the referee for this idea. We have added the advantages and disadvantages of the cell membrane fragmentation methods as a table (Table 3) at the end of part 4. We have done the same for the coating methods of part 7 (Table 6). In the case of the membrane fragments isolation methods in part 5, we have added a paragraph stating the comparations of the different centrifugation options.
- Part 10 (conclusions) can be deleted since the contents are repeated with part 8 and 9.
Reply to referee.
We agree with the referee and now this part has been deleted
Minor:
- In the title, the statement of “for tissue specific therapeutics” can be deleted since the paper mainly focuses on the coating technique.
Reply to referee.
We appreciate the reviewer's comment. However, with the reviewer's permission, we would like to maintain the current title, as the majority of particle targeting systems are envisioned for potential future therapeutic applications.
- In introduction part, I’d suggest citing the following 3 fresh and comprehensive papers to instead ref 1-9 efficiently, since they include all the advantages of nanomedicines and nanoencapsulations the authors mentioned:
https://www.sciencedirect.com/science/article/pii/S1748013218306388?via%3Dihub
https://www.sciencedirect.com/science/article/pii/S0378517318300784?via%3Dihub
https://pubs.acs.org/doi/full/10.1021/acsanm.3c04487
Reply to referee.
We thank the referee for this suggestion. Now these three new references have been included in the paper.
- What’s the unique advantage of membrane-coated NPs compared to other similar components such as EV/lipid-coated NPs? As far as I know, there are some reports that EV can also target cells. It’s better to state this in the introduction part of manuscript.
Reply to referee.
We thank the referee for the comment. We expand the introduction to underline the crucial aspect of targeting specific cells, which is a distinctive property of the system. Furthermore, Table 1 clearly describes the different types of cancer donor cells that are used to target their homologous cells. This information provides a more thorough overview of the advantages associated with membrane-coated NPs.
- It’s better to add the exact cell line in Table 1 as a new row.
Reply to referee.
We thank the referee for the comment. Table 1 has been changed.
- In Table 4, the corresponding NPs should also be listed as a row or combine it with Table 3 together.
Reply to referee.
We thank the referee for this idea. Table 4 (now Table 5) has now a new row with this information.
- In Figure 3, I’d suggest selecting more TOC/Scheme figures from other references you referred to (line 271-283) to replace Figure 3b-3e to show more interesting works to readers. Since you also didn’t talk in detail about these figures (3b-3e) in the text.
Reply to referee.
We thank the referee for this idea. We agree with the reviewer on the idea of putting more Schemes instead of results since they are more visual and understandable. Now Figure 3 has been changed and the schemes of nanoparticles coated with cell membranes of immune (a) or multiple myeloma cells (b) are provided.
- In part 9, I believe another direction is to increase the efficiency of membrane isolation and coating rate for NPs since both are too low at present.
Reply to referee.
Unfortunately, exact efficiency data is not available; however, all the information regarding efficiency and coating rates found in the papers has been included in the discussion section
Reviewer 3 Report
Comments and Suggestions for Authors
The manuscript submitted by Fernández-Borbolla et al. review the recent papers concerning the coating of different types of particles with cell membranes. The manuscript is clear, well organized and the authors have also proposed some interesting perspectives. However, I have some suggestions:
1. first of all, the authors must discuss about the coating mechanism. Which type of interactions appear between the particles and cell membranes?
2. more details about the influence of the coating on the particles size and stability, drug release kinetics must be provided.
3. cellular uptake studies have demonstrated that these functionalized particles are better than not coated ones?
Author Response
Manuscript Number: ijms-2785833
REVIEWER #3
The manuscript submitted by Fernández-Borbolla et al. review the recent papers concerning the coating of different types of particles with cell membranes. The manuscript is clear, well organized and the authors have also proposed some interesting perspectives. However, I have some suggestions:
- first of all, the authors must discuss about the coating mechanism. Which type of interactions appear between the particles and cell membranes?
Reply to referee.
We agree with the reviewer and therefore a paragraph has been added to the discussion explaining the coating mechanism.
- more details about the influence of the coating on the particles size and stability, drug release kinetics must be provided.
Reply to referee.
We consider this topic to be very important; therefore, a dedicated section has been included in the discussion part
- cellular uptake studies have demonstrated that these functionalized particles are better than not coated ones?
Reply to referee.
Yes, we believe this issue is crucial to clarify, as highlighted by the referee's comments, indicating that functionalized nanoparticles are better than non-functionalized ones. Empirical data supporting this assertion have been incorporated into the discussion.
Reviewer 4 Report
Comments and Suggestions for Authors
The manuscript by Fernández-Borbolla et al. “Cell Membrane-Coated Nanoparticles for Precision Medicine: A Comprehensive Review of Coating Techniques for Tissue-Specific Therapeutics” demonstrated the recent updates on cell membrane-coated nanoparticles synthesis, application, and perspectives. Overall, this manuscript requires revision before publication in IJMS as follows:
Comments
1. The abstract should be crispier and precisely revised.
2. Lines 31-38, the information can be elaborating in view of nanoparticle's properties (size, structure, porous nature, surface area, and functional groups on their surface as key roles for applications apart from their toxic nature), etc. Polymers 2022, 14(7), 1409.
3. Lines 62-67, this study's objectives and significance can be clearly stated.
4. Table 1, the descriptive details of the application should be added, i.e., quantitative data/results details. Also, please do the needful for Table 3 and Table 4.
5. In Table 2, please include the advantages and disadvantages of these buffers.
6. In Table 4, please provide the amount of bioactive loading.
7. Each section is minor polished with finding details (more quantitative information) of crucial citations.
8. The discussion can be more elaborated.
Comments on the Quality of English LanguageMinor editing of the English language is required.
Author Response
Manuscript Number: ijms-2785833
REVIEWER #4
The manuscript by Fernández-Borbolla et al. “Cell Membrane-Coated Nanoparticles for Precision Medicine: A Comprehensive Review of Coating Techniques for Tissue-Specific Therapeutics” demonstrated the recent updates on cell membrane-coated nanoparticles synthesis, application, and perspectives. Overall, this manuscript requires revision before publication in IJMS as follows:
Comments
- The abstract should be crispier and precisely revised.
Reply to referee.
The abstract has been rewritten following the reviewer's suggestions.
- Lines 31-38, the information can be elaborating in view of nanoparticle's properties (size, structure, porous nature, surface area, and functional groups on their surface as key roles for applications apart from their toxic nature), etc. Polymers 2022, 14(7), 1409.
Reply to referee.
We thank the referee for this idea. The introduction has been changed.
- Lines 62-67, this study's objectives and significance can be clearly stated.
Reply to referee.
Many thanks to the referee for his comments. As per his/her suggestions, we have revised this paragraph to elucidate the objectives and significance of our work.
- Table 1, the descriptive details of the application should be added, i.e., quantitative data/results details. Also, please do the needful for Table 3 and Table 4.
Reply to referee.
We thank the referee for the comment. However, we respectfully feel that incorporating all that information into this table may be excessive, and it may not align with the specific purpose we have envisioned for this context.
- In Table 2, please include the advantages and disadvantages of these buffers.
Reply to referee.
Thank you for your suggestion. Unfortunately, the papers we referenced do not provide explicit information on the advantages and disadvantages of the buffers.
- In Table 4, please provide the amount of bioactive loading.
Reply to referee.
We thank the referee for this idea. The amount of bioactive loading has now been included.
- Each section is minor polished with finding details (more quantitative information) of crucial citations.
Reply to referee.
We agree with the referee's opinion and the paper has been greatly reorganized (all changes are highlighted in yellow). Furthermore, more than 45 references have been included.
- The discussion can be more elaborated.
Reply to referee.
We agree with the referee and now the discussion has been carefully expanded
Reviewer 5 Report
Comments and Suggestions for Authors
The work entitled “Cell Membrane-Coated Nanoparticles for Precision Medicine: A Comprehensive Review of Coating Techniques for Tissue-Specific Therapeutics” reports on the ongoing developments and approaches to cell membrane-coated nanoparticles that position this technology as a promising alternative for effective targeted drug delivery and many other therapeutic applications.
The work is very well organized, put together and simple to follow. The authors did not exceed their observations in the presentation of the researches but they should present in which way they did the selection of the presented data. For instance, how many researches were in their initial focus group, how were they reduced, etc. basically the methodology employed for this work. Considering, they organized this review as if a regular paper was presented this is required. Also, the discussion is lacking a bit depth. The authors are discussing the collected information a bit superficially and this is such an interesting subject that deserves a more in depth analysis. The novelty of this review work and the need for it is also not highlighted. This should be fixed as well.
Comments on the Quality of English LanguageThere are grammar and writing errors along the manuscript that should be fixed.
Author Response
Manuscript Number: ijms-2785833
REVIEWER #5
The work entitled “Cell Membrane-Coated Nanoparticles for Precision Medicine: A Comprehensive Review of Coating Techniques for Tissue-Specific Therapeutics” reports on the ongoing developments and approaches to cell membrane-coated nanoparticles that position this technology as a promising alternative for effective targeted drug delivery and many other therapeutic applications.
The work is very well organized, put together and simple to follow.
We express our gratitude to the reviewer for her/his positive comments and for finding our work interesting.
The authors did not exceed their observations in the presentation of the researches but they should present in which way they did the selection of the presented data. For instance, how many researches were in their initial focus group, how were they reduced, etc. basically the methodology employed for this work. Considering, they organized this review as if a regular paper was presented this is required.
Reply to referee.
At the outset, a few relevant articles were identified through a search on Google Scholar focusing on cell membrane-coated nanoparticles. Subsequently, additional articles referencing these initial selections were included. To further expand our understanding, a subsequent search was conducted specifically targeting articles discussing the coating procedures of these nanoparticles.
Following the feedback from the initial review round, there was a request for more quantitative information. In response, we conducted another search to include more analytically detailed articles. Initially, we restricted the search to articles published up to 2020 to focus on recent advancements. However, we later refined the search parameters to include only articles from 2023 and 2024, thereby enriching the review with the most up-to-date knowledge in this field
Also, the discussion is lacking a bit depth. The authors are discussing the collected information a bit superficially and this is such an interesting subject that deserves a more in depth analysis.
Reply to referee.
We agree with the reviewer, and since the discussion is central to this paper, we have implemented it considerably (changes are highlighted in yellow).
The novelty of this review work and the need for it is also not highlighted. This should be fixed as well.
Reply to referee.
Following the referee´s suggestions, we have added a specific paragraph at the end of the introduction to elucidate the objectives and significance of our work
Round 2
Reviewer 1 Report
Comments and Suggestions for Authors
The revisions have addressed most of the issues. However, for the background of precise therapy and medicine, the related advances should be added for better reviewing including: 10.1021/acsami.1c16859, 10.1002/EXP.20230025.
Reviewer 4 Report
Comments and Suggestions for Authors
Accept
Reviewer 5 Report
Comments and Suggestions for Authors
The authors have implemented all the recommended alterations. The manuscript is now ready for publication.